
**The effect of COVID-19 restrictions on atmospheric new particle formation**
**in Beijing**
Chao Yan[1,2,#], Yicheng Shen[3,#], Dominik Stolzenburg[2], Lubna Dada[2,4], Ximeng Qi[5], Simo
Hakala[2], Anu-Maija Sundström[6], Yishuo Guo[1], Antti Lipponen[7], Tom V. Kokkonen[5], Jenni
Kontkanen[2], Runlong Cai[2,3], Jing Cai[1,2], Tommy Chan[2], Liangduo Chen[5], Biwu Chu[2], Chenjuan
Deng[3], Wei Du[1,2], Xiaolong Fan[1], Xu-Cheng He[2] , Juha Kangasluoma[1,2], Joni Kujansuu[1,2],
Mona Kurppa[2], Chang Li[1], Yiran Li[3], Zhuohui Lin[1], Yiliang Liu[8], Yuliang Liu[5], Yiqun Lu[8],
Wei Nie[5], Jouni Pulliainen[6], Xiaohui Qiao[3], Yonghong Wang[1,2], Yifan Wen[3], Ye Wu[3], Gan
Yang[8], Lei Yao[2], Rujing Yin[3], Gen Zhang[9], Shaojun Zhang[3], Feixue Zheng[1], Ying Zhou[1], Antti
Arola[7], Johanna Tamminen[6], Pauli Paasonen[2], Yele Sun[10], Lin Wang[8], Neil M. Donahue[11],
Yongchun Liu[1], Federico Bianchi[2], Kaspar R. Daellenbach[2,4], Douglas R. Worsnop[2,12], Veli-
Matti Kerminen[2], Tuukka Petäjä[2,5], Aijun Ding[5,*], Jingkun Jiang[3,*], Markku Kulmala[1,2,5*]
**Affiliations:**
[1] Aerosol and Haze Laboratory, Beijing Advanced Innovation Center for Soft Matter Science and Engineering,
Beijing University of Chemical Technology, Beijing, China
[2] Institute for Atmospheric and Earth System Research / Physics, Faculty of Science, University of Helsinki,
Finland
[3] State Key Joint Laboratory of Environment Simulation and Pollution Control, State Environmental Protection
Key Laboratory of Sources and Control of Air Pollution Complex, School of Environment, Tsinghua University,
Beijing, China
[4] Laboratory of Atmospheric Chemistry, Paul Scherrer Institute, 5232 Villigen, Switzerland.
[5] Joint International research Laboratory of Atmospheric and Earth System Research (JirLATEST), School of
Atmospheric Sciences, Nanjing University, Nanjing, China.
[6] Finnish Meteorological Institute, 00560 Helsinki, Finland
[7] Finnish Meteorological Institute, 70211 Kuopio, Finland
[8] Department of Environmental Science & Engineering, Fudan University, Shanghai, China
[9] State Key Laboratory of Severe Weather & Key Laboratory of Atmospheric Chemistry of China Meteorological
Administration (CMA), Chinese Academy of Meteorological Sciences, Beijing 100081, China
[10] Institute of Atmospheric Physics, Chinese Academy of Science, Beijing, China
[11] Center for Atmospheric Particle Studies, Carnegie Mellon University, Pittsburgh, PA, USA
[12] Aerodyne Research Inc., Billerica, Massachusetts 01821, USA
[#] these authors contributed equally to this work
[*] Correspondence to:
Markku Kulmala, markku.kulmala@helsinki.fi
Jingkun Jiang, jiangjk@tsinghua.edu.cn
Aijun Ding, dingaj@nju.edu.cn

**Abstract**
During the COVID-19 lockdown, the dramatic reduction of anthropogenic emissions provided
a unique opportunity to investigate the effects of reduced anthropogenic activity and primary
emissions on atmospheric chemical processes and the consequent formation of secondary
pollutants. Here, we utilize comprehensive observations to examine the response of
atmospheric new particle formation (NPF) to the changes in the atmospheric chemical cocktail.
We find that the main clustering process was unaffected by the drastically reduced traffic
emissions, and the formation rate of 1.5 nm particles remained unaltered. However, particle
survival probability was enhanced due to an increased particle growth rate (GR) during the





lockdown period, explaining the enhanced NPF activity in earlier studies. For GR at 1.5–3 nm,
sulfuric acid (SA) was the main contributor at high temperatures, whilst there were
unaccounted contributing vapors at low temperatures. For GR at 3–7 nm and 7–15 nm,
oxygenated organic molecules (OOMs) played a major role. Surprisingly, OOM composition
and volatility were insensitive to the large change of atmospheric $NO_x$ concentration; instead
the associated high particle growth rates and high OOM concentration during the lockdown
period were mostly caused by the enhanced atmospheric oxidative capacity. Overall, our
findings suggest a limited role of traffic emissions in NPF.

## 58    1. Introduction

The pandemic of COVID-19 has led to the death of more than 5.3 million individuals
globally [WHO 2020, *https://covid19.who.int/*]. Restrictions on population movement
(lockdowns) worldwide led to arguably the most significant reduction of primary
anthropogenic emissions in recent history. $NO_x$ concentrations declined on average by about
50 – 60 % in several European, South American, Indian, and Chinese cities (Sicard et al.,
2020;Krecl et al., 2020;Shi and Brasseur, 2020;Agarwal et al., 2020), and mixing ratios of other
primary pollutants, such as black carbon (BC), carbon monoxide (CO), sulfur dioxide ($SO_2$),
and volatile organic compounds (VOCs) were also reduced in varying degrees (Bao and Zhang,
2020; Chu et al., 2021; Shen et al., 2021b; Xing et al., 2020; Pei et al., 2020).
The reductions of primary emissions mitigated particulate pollution and improved air quality
in many countries around the globe (Sicard et al., 2020; Krecl et al., 2020; Agarwal et al., 2020;
Ciarelli et al., 2021), including many Chinese cities (Wang et al., 2020b; Huang et al., 2021;
Le et al., 2020). However, the reduction of $PM_{2.5}$ was considerably weaker than those of the
primary pollutants, and in some cities such as Beijing, the $PM_{2.5}$ concentrations even increased
after the lockdown policy was imposed (Huang et al., 2021). This persistent particulate
pollution has been attributed to both unfavorable meteorology, such as stagnant meteorological
conditions and high relative humidity (RH) (Le et al., 2020; Wang et al., 2020b) and to
enhanced atmospheric oxidative capacity caused by increased $O_3$ and $NO_3$ radical formation
(Huang et al., 2021;Le et al., 2020). To date, few studies have focused on either atmospheric
new particle formation or the overall particle number size distribution (Shen et al., 2021a; Shen
et al., 2021b) during the lockdown period, although NPF has been shown to enhance haze
formation (Guo et al., 2014; Kulmala et al., 2021), and the particle number size distribution is
known to influence the health effect of particles (Harrison et al., 2010).



NPF contains two consecutive stages: formation of particles via molecular clustering followed
by particle growth (Kulmala et al., 2014). A complete understanding of both stages remains
elusive in polluted urban environments. In the first stage, a key concern is the identity of the
clustering molecules. On one hand, several laboratory studies (Almeida et al., 2013; Xiao et
al., 2021) and ambient measurements (Yao et al., 2018; Yin et al., 2021; Yan et al., 2021; Cai
et al., 2021b; Deng et al., 2020) indicate that clustering between sulfuric acid (SA) and amines
drives the initial NPF in polluted environments. On the other hand, there are also studies
suggesting that organic acids formed from oxidation of traffic emissions are key clustering
species (Guo et al., 2020). The contrast between the enhanced NPF and reduced traffic load
during the lockdown period seems to support the former mechanism, but a detailed
investigation of how molecular clustering responded to those emission reductions remains
lacking. For the growth phase, oxygenated organic molecules (OOMs) have been shown to
dominate in some cases (Yan et al., 2021; Qiao et al., 2021). Further, a high fraction of nitrogen
containing OOMs suggests that $RO_2+NO_x$ reactions prevail in OOM formation (Qiao et al.,
2021). For monoterpene-derived OOMs, which is characteristic of a remote atmosphere, high
$NO_x$ levels can suppress particle growth by altering a fraction of products to organic nitrates
with higher volatilities (Yan et al., 2020). However, the effect of $NO_x$ in OOM formation and
particle growth needs to be examined in urban settings, where the VOC precursors are largely
different.
Enhanced NPF during the lockdown period has been reported (Shen et al., 2021b), but without
a detailed explanation due to the lack of simultaneous measurements of both particles at the
size where NPF starts (e.g., 1.5 nm) and key vapors for NPF, such as SA and OOMs. We fill
that gap with comprehensive measurements from urban Beijing covering the lockdown period,
enabling the investigation on how NPF responded to the emission reductions during lockdown
on molecular and process levels.
**2.  Methodology**
*2.1 Measurement location and period*

The measurement campaign was conducted at the Aerosol and Haze Laboratory located at
the west campus of Beijing University of Chemical Technology (BUCT station, Lat. 39º56′31″
and Lon. 116º17′52″). It is a representative urban station surrounded by residential and
commercial areas and three main roads with heavy traffic loads. Measurements of atmospheric



variables and pollutants have been conducted continuously in this station since early 2018.
More details about the station and measurements can be found elsewhere (Liu et al., 2020).

The main data sets analyzed in this study were collected during 2019/12/15 – 2020/3/15,

divided into pre-lockdown (2019/12/15 - 2020/01/22) and lockdown (2020/01/23 – 2020/03/15)
periods. The Chinese Spring Festival (CSF) overlapped the lockdown period, but since they
have a similar effect on population movement, the CSF and COVID-19 periods were not
further separated in this study. As shown in Fig. S1-S3, the traffic congestion index, as well as
the $NO_2$ concentration measured by 11 national monitoring stations in Beijing and by satellite,
showed an apparent reduction and a slow rebound after the lockdown was imposed. In contrast,
traffic and the $NO_2$ concentration quickly rebounded after the CSF in 2019.

*2.2 Instrumentation*

The particle number size distribution over in the diameter range of 1 nm - 10 μm was

measured by the combination of a diethylene glycol scanning mobility particle spectrometer
(DEG-SMPS, 1-7.5 nm) and a particle size distribution system (PSD, 3 nm-10 μm).  Particle
formation rates ($J_{1.5}$, $J_3$, $J_6$, and $J_{10}$) were calculated for all NPF cases using a new balance
formula that is optimized for polluted environments (Cai and Jiang, 2017). NPF is classified
according to the commonly-used criteria originally described by Dal Maso and co-workers
(Dal Maso et al., 2005), i.e., 1) a burst of sub-3 nm particles, and 2) continuous particle growth
in size. In some cases when only criterium 1 is satisfied, referred to as clustering events
hereafter, particle formation rates can still be calculated because growth is negligible compared
to coagulation and the dN/dt term for such small particles is in the formula. Hence, we included
both NPF events and clustering events when we investigated $J_{1.5}$, $J_3$, and their response to other
relevant parameters. In addition, the condensation sink (CS) was calculated based on the
measurement of particle number size distribution (Kulmala et al., 2012). Detailed calculations
of particle formation rate and growth rate are provided in Supplementary Information (SI).

SA and OOM concentrations were measured with a nitrate ion-based Chemical Ionization

Atmospheric-Pressure-interface Long-Time-of-Flight mass spectrometer (CI-APi-LTOF,
Aerodyne Research, Inc.). The configuration of this instrument has been described previously
(Yan et al., 2021). Two levels of calibrations were performed. First, the SA concentration was
calibrated following the same procedure suggested by Kürten et al., (2012); Second, the mass-
dependent transmission efficiency of the instrument was obtained with the method developed


by Heintrizi et al., (2016). After these calibrations, the concentration of SA and OOMs can be
calculated using the equations below:
$[SA] = \frac{HSO_4^- + (HNO_3)HSO_4^-}{\sum_{i=0}^{2}(HNO_3)_i NO_3^-} \times C$ Eq. (1)
$[OOM] = \frac{(OOM)NO_3^- + (OOM-H)^-}{\sum_{i=0}^{2}(HNO_3)_i NO_3^-} \times C \div T_{OOM}$ Eq. (2)
In the righthand side of Eqs. 1 and 2, the numerator and denominator are the signals of analytes
and reagent ions, respectively, and C denotes the calibration coefficient obtained for SA, which
was determined as $7.0 \times 10^9$ (molecule·cm$^{-3}$)/ncps. $T_{OOM}$ in Eq.2 is the mass-dependent
transmission efficiency relative to the reagent ions.

In addition, we measured the concentrations of CO, $SO_2$, $NO_x$, and $O_3$ using four Thermo

Environmental Instruments (models 48i, 43i-TLE, 42i, 49i, respectively). These trace-gas
pollutants were sampled through a 3-meter tube from the building roof, which was heated to
313 K to reduce sampling losses. Calibrations of these instruments were performed bi-weekly
using standard gases of known concentrations. In addition, several meteorological varibles,
including the ambient temperature, relative humidity, pressure, visibility, UVB radiation, as
well as horizontal wind speed and direction, were measured with a weather station (AWS310,
Vaisala Inc.) located on the rooftop of the building. More details of these instruments are
provided in SI.
**3. Results and discussion**
*3.1 Changes of atmospheric pollutants during the pre-lockdown and the lockdown periods*

We first investigated the extent to which lockdown restrictions modified pollution

concentrations.  Figure 1 is an overview of the particle number size distributions and some
other relevant pollutants. As shown in Figure 1A&B, NPF occurred more frequently in the
lockdown period (30.8 %, 16 out of 52 days) than in the pre-lockdown period (18.0 %, 7 out
of 39 days). This difference is reduced if we include clustering events, to 34.6 % and 28.2 %
for the lockdown and pre-lockdown periods, respectively. Consistent with a few recent studies
(Cai et al., 2017; Yan et al., 2021; Deng et al., 2021), the burst in the concentration of sub-3
nm particles ($N_{1-3}$ in Figure 1B) corresponded to a low CS during both periods, suggesting that
CS was the governing parameter for NPF.

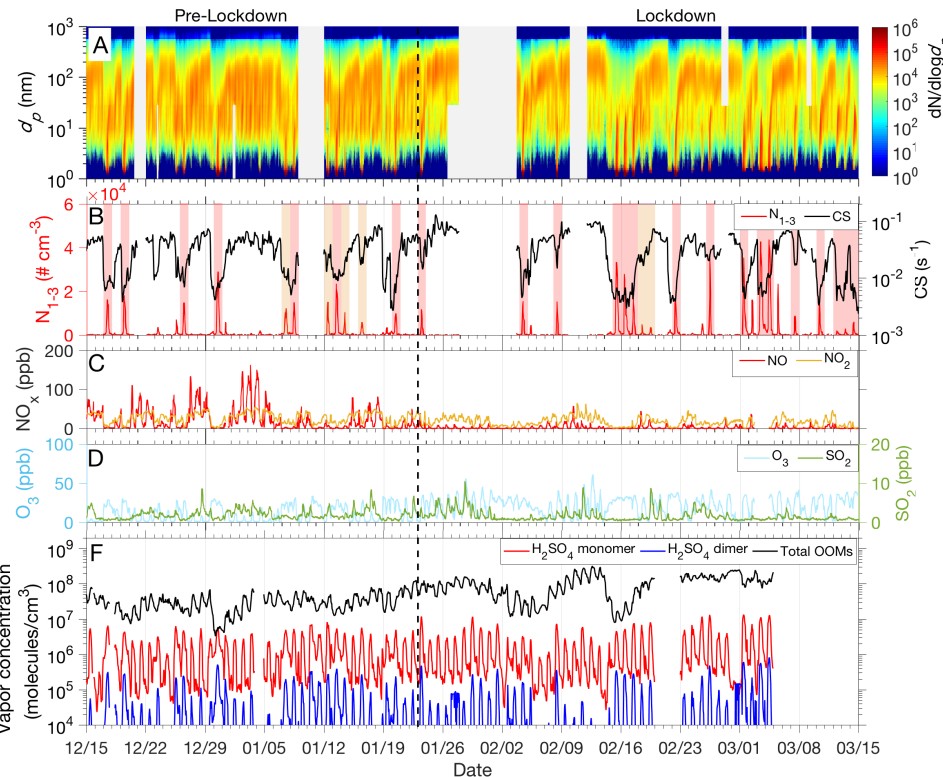


**Figure 1.** Concentrations of atmospheric pollutants during the pre-lockdown and lockdown periods,
including the particle number size distribution (A), number concentration of 1.5-3 nm particles ($N_{1-3}$)
and NPF classification (B), NO and $NO_2$ (C), $SO_2$ and $O_3$ (D), and $H_2SO_4$ monomer, dimer, and total
oxygenated organic molecules (OOMs) (F). The vertical dashed line denotes the separation of the pre-
lockdown and the lockdown periods. In Panel B, days with NPF events and clustering events are shaded
in red and orange, respectively.

One prominent change of the particle number size distribution during the lockdown was
that particles in the size range of 10-30 nm were significantly reduced during the traffic rush
hours (Fig. S4), indicating that vehicle emissions contributed substantially to particles of this
size range during this time window. However, particles below this size range were not
substantially depleted, indicating a limited contribution of traffic emissions to the sub-10 nm
particle concentration in Beijing. This is in contrast to findings in some European countries
(Ronkko et al., 2017). A likely reason for this difference is that the observed NPF in Beijing
was notably more intense than in European cities, so it may have overwhelmed the contribution
of traffic emissions. Also, the abundant background aerosols in Beijing may have scavenged
any freshly-emitted particles more efficiently, so that fewer of these primary nano-particles
survived until observation.





Significant changes in the concentration of trace-gas pollutants coincided with the

lockdown. As shown in Figure 2A-C, NO, $NO_2$, and $SO_2$ concentrations during NPF periods
(7 am – 6 pm) decreased by 3.2-, 2.0-, and 3.0-fold (median values), respectively. As mentioned
above, the reduction of $NO_x$ ($NO_x=NO+NO_2$) was directly related to the restriction of traffic.
However, the reduced $SO_2$ concentration was likely unrelated to the traffic restriction, because
the $SO_2$ concentrations did not exhibit a typical traffic pattern in either the pre-lockdown or the
lockdown period. Unlike the primary pollutants, $O_3$ concentrations increased by 25 % (Figure
2D), consistent with previous studies (Huang et al., 2021). Moreover, in comparison to the pre-
lockdown period, temperature and UVB radiation were higher during the lockdown period (Fig.
S5), suggesting stronger atmospheric photochemistry.

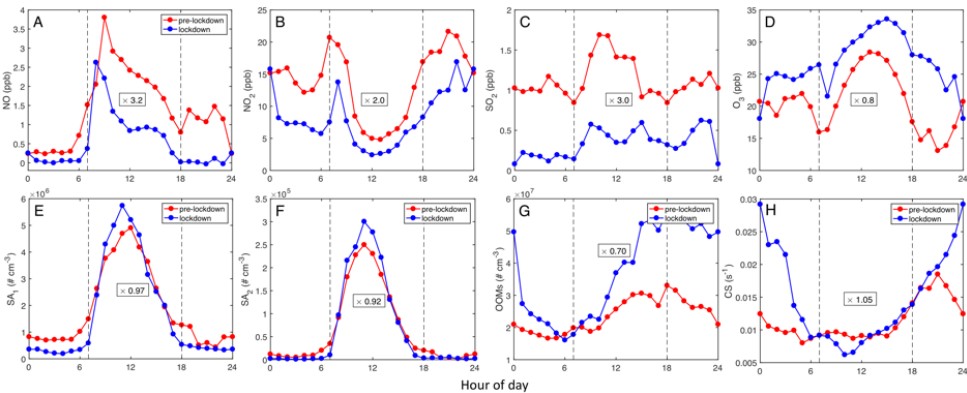


**Figure 2**. Median diurnal cycles of atmospheric variables during the pre-lockdown and lockdown
periods, including NO, $NO_2$, $SO_2$, $O_3$, $SA_1$, $SA_2$, OOMs, and CS. The ratio of $[X]_{pre\text{-}lockdown}/[X]_{lockdown}$ is
given in the framed text. Here, [X] denotes the average value of a specific atmospheric variables during
the NPF time window, i.e., 7am – 6 pm, as marked by the two dashed lines.

The corresponding changes in the most NPF-relevant parameters, including sulfuric acid

monomers ($SA_1$), dimers ($SA_2$), oxygenated organic molecules (OOMs) and CS, are shown in
Figures 2E-H. The CS was almost identical between the pre-lockdown and lockdown periods
(Figure 2H). The median $SA_1$ and $SA_2$ concentrations were also stable between the two periods.
This is because the decline of the sulfuric acid precursor (i.e., $SO_2$, Figure 2C) was completely
compensated by the enhanced photochemistry, as indicated by the variation of UVB (Fig. S5B).
In addition, the concentration of OOMs increased by about 50% during the lockdown. This is
because the concentration of volatile organic compounds (VOCs) only declined slightly in the
lockdown period (Shen et al., 2021b), but the photochemistry was much more enhanced.
***3.2 Changes in initial particle formation rate and size-segregated growth rates***



Based on our previous studies of the governing factors and mechanism of NPF in Beijing
(Cai et al., 2017; Yan et al., 2021; Deng et al., 2021), we would expect the formation rates of
1.5 nm particles $(J_{1.5})$ during the two periods to be very similar because $SA_1$, $SA_2$, and CS were
nearly identical. However, this was not the case; a previous study in Beijing showed that NPF
was more intense during the lockdown period than in the pre-lockdown period (Shen et al.,
2021b). In order to resolve this puzzle, we examined the detailed formation rates calculated for
particles of different sizes, i.e., $J_{1.5}$, $J_3$, $J_6$, and $J_{10}$. We compare these formation rates in Figure
3A. Consistent with our initial expectation, $J_{1.5}$ was very similar in these two periods; however,
at progressively larger particle sizes the difference of particle formation rates during the two
periods becomes progressively more pronounced. This means that, while the nucleation rates
remained constant, more of the newly formed particles survived during the lockdown period.
As shown in Figure 3B, the particle survival probabilities, calculated as $J_{dp2}/J_{dp1}$ from 1.5 nm
to 3 nm, 6 nm, and 10 nm during the lockdown period were enhanced by factors of 1.2, 1.9 and
4.4, respectively, compared to pre-lockdown conditions. This provides one explanation for the
enhanced particle formation rates reported previously – if the particles were only measured at
a size larger than 1.5 nm, the calculated formation rate would be larger in the lockdown period
due to the enhanced particle survival probability. In addition, despite the similar median values
of $J_{1.5}$, a few intense NPF cases occurred during the lockdown period, in contrast to the pre-
lockdown period (Figure 3C). In such cases, the classification of NPF events, which is to some
extent subjective, could also affect the comparison (a classification bias). For instance, if weak
NPF events were not detected or counted, the average $J_{1.5}$ during the lockdown period would
be higher. This could be another reason for the reported stronger NPF in the lockdown period
(Shen et al., 2021b).

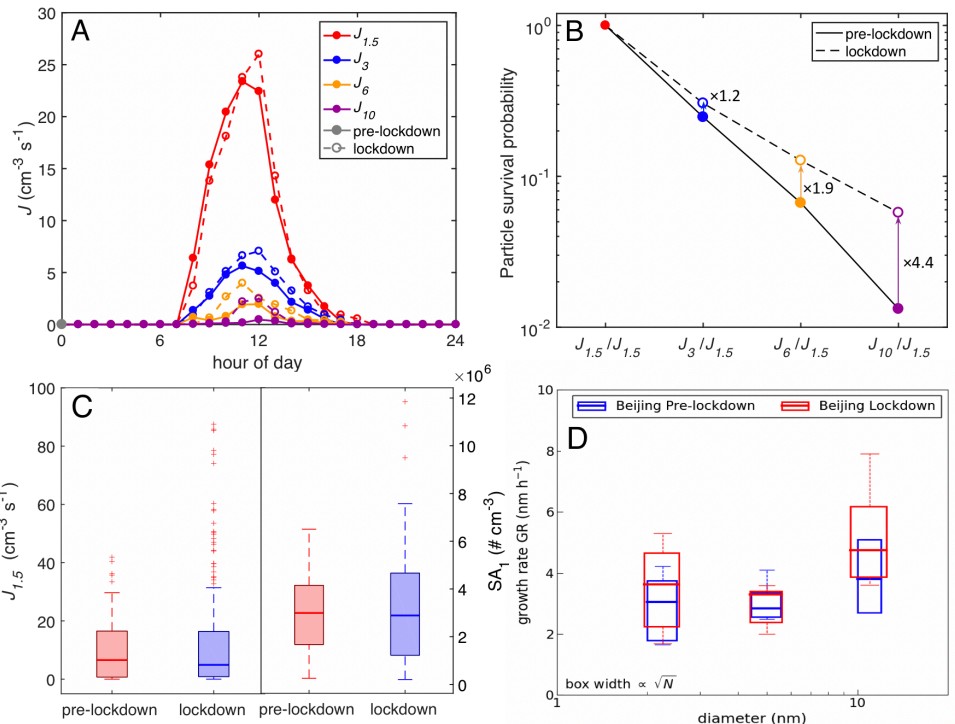


**Figure 3**. The diurnal cycles of particle formation rates, growth rates, and survival probability (median values) at different sizes during the pre-lockdown and lockdown periods. (A) diurnal variations of particle formation rates at different sizes, i.e., $J_1$, $J_3$, $J_6$ and $J_{10}$. (B) Particle survival probability as a function of size. (C) Box plots showing the distribution of $J_{1.5}$ and $SA_1$. (D) Size-segregated particle growth rates.

The particle survival probability is mostly determined by the competition between particle growth and scavenging by pre-existing large particles (Kerminen and Kulmala, 2002; Lehtinen et al., 2007). As the scavenging rate of nanoparticles is approximately proportional to CS, the particle survival probability is proportional to the ratio of particle growth rate (GR) to CS (GR/CS) (Kulmala et al., 2017). In our observations, CS values during the time windows of NPF events were similar in these two periods (Figure 2H), so a change in GR must be the key to the different particle survival probability. To explore this, we calculated size-dependent growth rates of sub-10 nm particles in the pre-lockdown and lockdown periods with the appearance-time method. This method gives a higher GR than the mode-fitting method (Deng et al., 2020; Qiao et al., 2021). Consistent with previous studies (Deng et al., 2020; Qiao et al., 2021), larger particles had higher growth rates (Figure 3D). The reason for the enhanced particle growth will be discussed in detail in Section 3.4.




### 3.3 Insights into the clustering mechanism and its response to the lockdown conditions


An important conclusion from our observations is that the clustering efficiency was not
significantly affected by the lockdown restrictions, as otherwise, $J_{1.5}$ would have most likely
changed drastically even though the SA concentration and CS were identical. For example, it
has been shown that particle formation rates differ by up to a factor of 1000 when SA clusters
with dimethylamine (DMA) instead of ammonia (Almeida et al., 2013), for constant SA and
CS. Hence, we further investigated the clustering efficiency of SA and the relationship between
$SA_2$ and $J_{1.5}$, focusing on comparisons between the pre-lockdown and lockdown periods.
An important diagnostic of SA clustering is the efficiency of $SA_2$ formation via the collision
of two $SA_1$. Here, $SA_1$ and $SA_2$ denote monomers and dimers of SA, which may also contain
base molecules acting as the stabilizer. Those base molecules cannot be seen by the nitrate-CI-
APi-TOF because of their evaporation during charging processes or inside the instrument
(Kurten et al., 2014). As the stabilizing effect of amines is much stronger than that of ammonia,
$SA_2$ formation efficiency is notably higher in the SA-amine system than in the SA-ammonia
system (Almeida et al., 2013; Kurten et al., 2014). In addition, the $SA_2$ formation efficiency
also depends on the concentration of base molecules, CS, as well as on the temperature (Cai et
al., 2021a). As shown in Figure 4, the most prominent feature of the $SA_2$ formation efficiency
in our observations is a clear dependence on temperature; $SA_2$ concentrations were consistently
lower at higher temperatures. This dependence was identical for both the pre-lockdown and
lockdown periods. On the other hand, the clustering efficiency appears to be independent of
CS, because the loss of $SA_1$-$DMA_1$ clusters was dominated by evaporation over the
temperature range of our observations (Fig. S6).
With a simplified SA-DMA clustering approach (Cai et al., 2021a), we were able to
reproduce the $SA_2$ formation, including its temperature dependence. Since the clustering
efficiency was not affected by CS, we set CS=0.01 s$^{-1}$ for the simulations. For that CS, the best
simulation result was obtained when the DMA concentration was constant at 1.3 ppt with a
50 % uncertainty, showing no systematic difference between the pre-lockdown and lockdown
periods (Fig. S7). This is less than the measured DMA concentration in 2018 in Beijing (Deng
et al., 2020). It should be noted that this effective DMA concentration (i.e., 1 ppt) is not the
"real" concentration of DMA, but rather it means that the stabilizing effect of all base
molecules is equivalent to that of 1 ppt DMA.

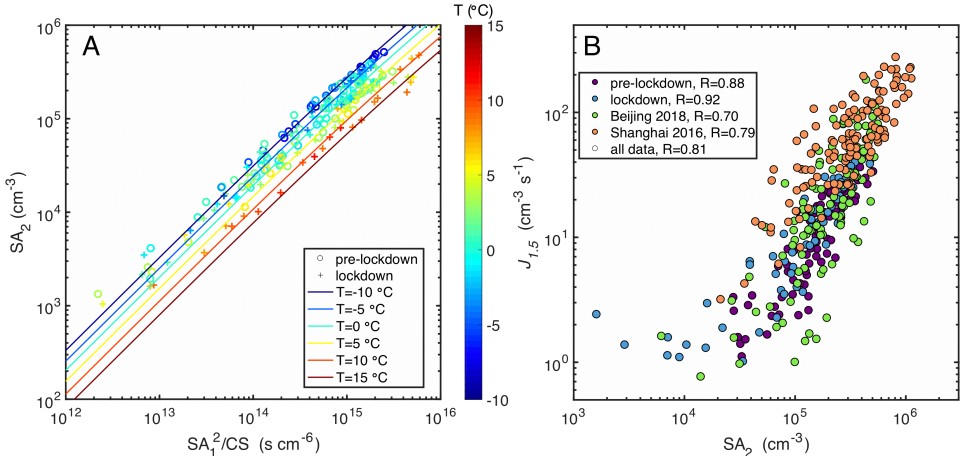

**Figure 4**. Clustering of SA and formation of new particles during the pre-lockdown and lockdown periods. (A) Measured daytime (7 am – 6 pm) $SA_2$ (dimer) concentration versus squared $SA_1$ concentration divided by CS, color-coded by temperature. This represents the dimer production efficiency. Lines denote clustering model simulations(Cai et al., 2021a). The simulations deployed a constant dimethylamine concentration (1 ppt) and CS (0.01 s$^{-1}$), which provided the best agreement with the ambient measurements. (B) Measured particle formation rate $J_{1.5}$ versus $SA_2$ concentration color-coded by different datasets. Measurements in 2018 wintertime Beijing (Yan et al., 2021) and in Shanghai (Yao et al., 2018) are also included. It should be noted that, $J_{1.7}$ was used in the study in Shanghai.

As shown in Figure 4B, $J_{1.5}$ correlates well with the $SA_2$ concentration, indicating that particle formation is driven by SA clustering processes. The relationship between $J_{1.5}$ and the $SA_2$ concentration agrees well with earlier observations in Beijing (Yan et al., 2021) and Shanghai (Yao et al., 2018), with a correlation coefficient of 0.81 for all data. However, in comparison to those earlier studies, $J_{1.5}$ in our observations is slightly lower, which could be attributed to the lower DMA concentration as discussed above. Most importantly, Figure 4A and 4B clearly indicate that the mechanisms of both SA clustering and initial particle formation remained the same in the pre-lockdown and lockdown periods, although a clear temperature effect can be seen. This gives direct evidence that the gaseous species emitted by traffic exhaust and their associated photochemical products alone are not the main source of either sulfuric acid or new particles in Beijing.

### 3.4 Characteristics of oxygenated organic molecules and the contribution to particle growth

Particle growth is key to particle survival, and subsequently the climate and health effects. Therefore, it is essential to understand the vapors responsible for particle growth, as well as the reason why particle growth was enhanced during the lockdown period, in spite of reduced



primary emissions. As the sulfuric acid concentration remained stable (Figure 2E) and it had a
minor contribution to the growth of particles larger than 3 nm (Deng et al., 2020; Qiao et al.,
2021), the enhanced particle growth rates were more likely associated with corresponding
changes of OOMs than sulfuric acid.
Recent studies have suggested that elevated $NO_x$ can suppress the formation of low-
volatility vapors by inhibiting the autoxidation of $RO_2$ radicals (Yan et al., 2020). Due to the
significant $NO_x$ reductions during the lockdown period, pronounced changes in OOM
composition were expected. Such changes were indeed observed for some OOMs. For instance,
as shown in Figure 5A, the ratio between two indicative compound categories varied
significantly as a function of NO. Here, the categories $C_{6-9}H_{7,9,11,13}O_6N$ and $C_{6-9}H_{8,10,12,14}O_5$ are
the termination products of bicyclic peroxy radicals originating from aromatics ($C_{6-9}H_{7,9,11,13}O_5$)
(Wang et al., 2017) formed through reactions with NO and $HO_2$, respectively. When the NO
concentration declined from the pre-lockdown period to the lockdown period, the ratio of $C_{6-9}H_{7,9,11,13}O_6N$ concentration to $C_{6-9}H_{8,10,12,14}O_5$ concentration decreased as well.
However, the majority of OOMs were insensitive to the declining NO. For example, the
ratio of categories $C_{6-9}H_{11,13,15,17}O_6N$ and $C_{6-9}H_{12,14,16,18}O_5$ did not depend on the NO
concentration (Figure 5B). These compounds are presumably termination products of $C_{6-9}H_{11,13,15,17}O_4$ radicals through reactions with NO and $HO_2$, respectively. They have a double
bond equivalent (DBE) of 1, suggesting that they originate from aliphatic rather than aromatic
precursors. Their $NO_x$ sensitivity differs from the OOMs derived from aromatics. It could be
that even at low $NO_x$ concentrations, the reaction with NO is necessary to form OOMs from
these peroxy radicals, as nitrogen-containing OOMs were consistently far more abundant than
nitrogen-free OOMs. Figure 5C and 5D also show that the overall nitrogen number of OOMs
did not depend on NO.
The overall OOM composition was surprisingly insensitive to changes in $NO_x$
concentrations. OOM chemical characteristics, i.e., the distributions of carbon number, oxygen
number, nitrogen number, hydrogen number, hydrogen-to-carbon ratio, and oxygen-to-carbon
ratio, remained almost identical during the two periods (Fig. S8). The stable OOM composition
indicates similar "intrinsic" (300 K) volatility distributions in the pre-lockdown and lockdown
periods, as shown in Figure S8. The mean temperatures were about 274 K and 280 K in these
periods, respectively (Figure S4A), and as a result, the ambient-temperature OOM volatilities
were both lower than the intrinsic values but similar to each other due to the small temperature
difference (Fig. S9). Therefore, we conclude that the influences of both temperature and
$RO_2+NO_x$ chemistry on OOM vapor condensation and the resulting particle growth rates were
very small. Though the OOM volatility distribution was stable between the pre-lockdown and
lockdown periods, the OOM concentrations increased during the lockdown period, likely due
to enhanced photochemistry.

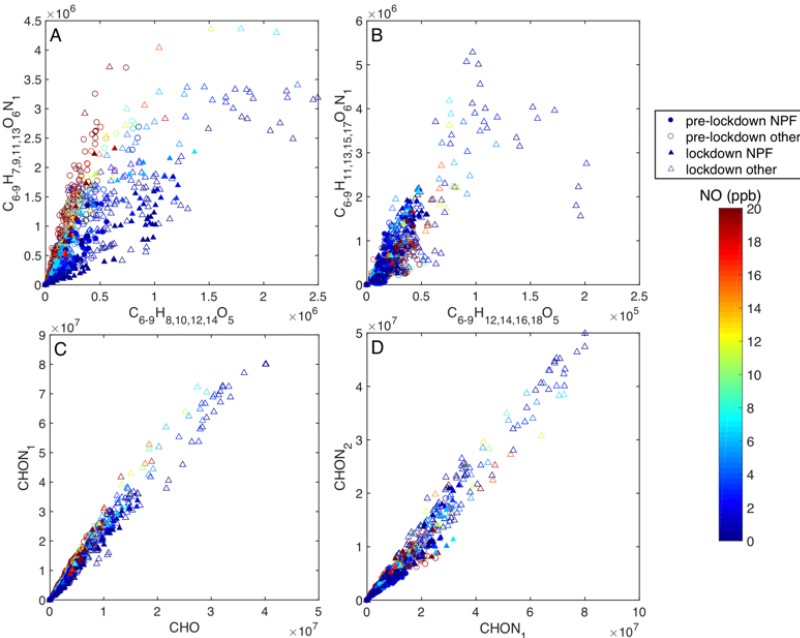


**Figure 5**. The influence of NO (given by symbol color) on the composition of OOMs, indicated by the
ratio between nitrogen-containing and nitrogen-free OOMs. (A) Selected OOMs with a double-bond-
equivalent (DBE) of 3, which are usually products from the oxidation of aromatic compounds(Molteni
et al., 2018;Wang et al., 2017;Garmash et al., 2020). (B) Selected OOMs with a DBE of 1, which are
more likely formed from the oxidation of aliphatic compounds, such as alkenes and alkanes. (C) OOMs
containing 0 and 1 nitrogen atom. (D) OOMs containing 1 and 2 nitrogen atoms. In all panels, only
daytime data (7:00 – 18:00) were included as they are directly relevant to NPF. Circles and triangles
represent data in pre-lockdown and lockdown periods, respectively; filled and empty markers denote
data during NPF days and other days, respectively.

Next, we examine contributions of SA and OOM to observed GRs in different size ranges,
i.e., 1.5 – 3 nm ($GR_{1.5-3}$), 3 – 7 nm ($GR_{3-7}$), and 7 – 15 nm ($GR_{7-15}$). Overall, this shows that
different processes govern growth at different sizes and temperatures.
Sulfuric acid contributed a relatively constant 1-1.5 nm/h to GR1.5-3 as shown in Figure
6A. At high temperatures (T > 0 °C) this explains most of the growth. However, at low
temperatures (T < 0 °C), SA condensation alone does not explain the observed $GR_{1-3}$,
suggesting an important contribution of other vapors favored by low temperatures. The vapors
and processes responsible for the residual $GR_{1.5-3}$ remain unclear, but they do not appear to be
OOMs, since the residual $GR_{1.5-3}$ ($GR_{measured} − GR_{SA}$) after subtracting SA contribution does





not show a positive correlation with condensable OOM concentration (Fig. S10). In fact, the
residual $GR_{1-3}$ shows a negative correlation with OOM concentration, mainly because of the
coincidence of high OOM concentration and high temperature. One possibility could be the
co-condensation of nitric acid and ammonia at low temperatures, as recently reported in
controlled chamber experiments (Wang et al., 2020a). However, observational evidence is
required to verify this hypothesis.

Above 3 nm, the growth rate from sulfuric acid condensation drops well below 1 nm/h, and

condensation from observed OOMs explains 1-4 nm/h of additional growth. For $GR_{3-7}$ the
OOM condensation correlates well with the observed GR (R=0.87) but the calculated GR was
lower than the observed value by roughly a factor of 2. The largest observed and calculated
growth was at the highest temperature during the lockdown, suggesting more efficient
photochemical production, though a residual excess at lower temperature may be related to
nitric acid and ammonia condensation. The correlation between calculated and observed
growth degrades for $GR_{7-15}$, though the highest observed and calculated values continue to be
at higher temperature. Given the growth rates, these particles are several hours old, and so
urban inhomogeneity may degrade this local analysis.
In laboratory experiments for growth by condensation of terpene oxidation products it has been
shown that the nitrate cluster ionization can miss up to half of the condensable organic vapors
(Trostl et al., 2016; Stolzenburg et al., 2018), and this could be true as well for these urban
conditions. If we scale the measured OOM concentrations with the same factors used by Tröstl
et al., (Trostl et al., 2016) (see SI), the measured and calculated GR fall close to the one-to-one
line (Fig. S10). Hence, underestimated OOM concentrations may well explain the under
predictions above 3 nm, although other possible reasons cannot be fully excluded, for instance,
the contribution of multiphase chemistry.




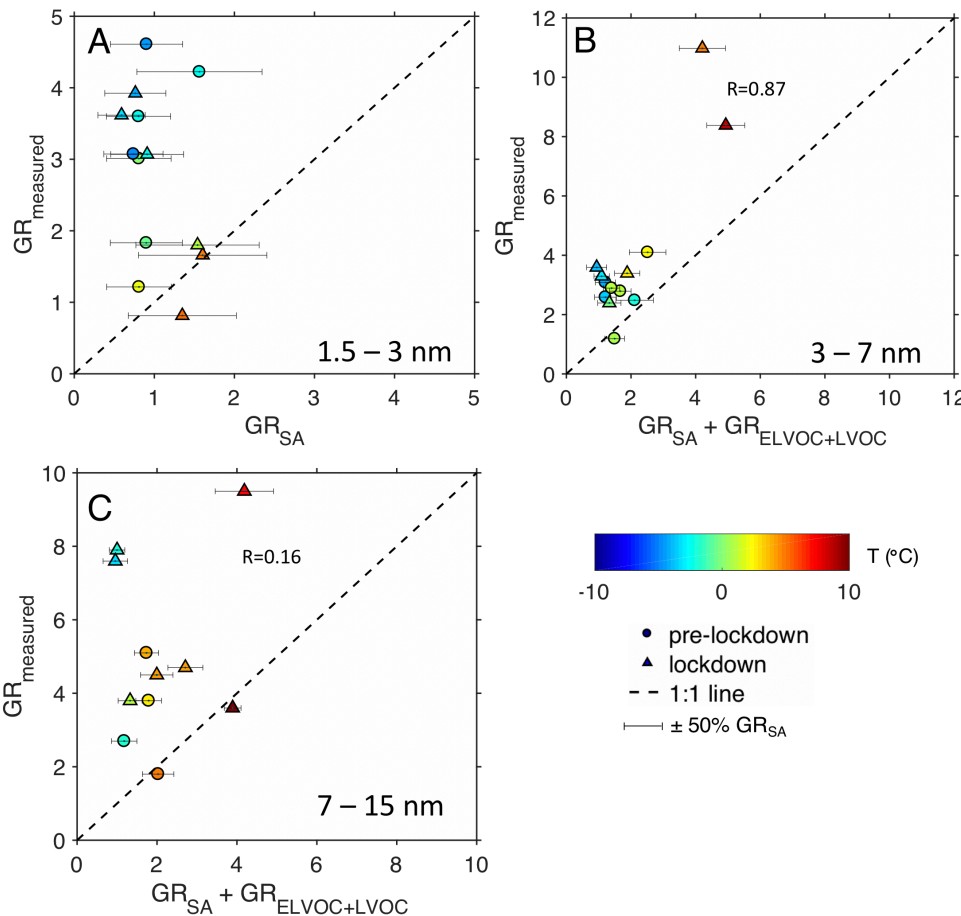


**Figure 6**. Connections of size-segregated particle growth rates to the plausible vapor concentrations. (A). The observed $GR_{1.5-3}$ versus the $GR_{1.5-3}$ predicted with SA. (B). $GR_{3-7}$ versus the predicted GR considering SA and condensable OOMs (the sum of ULVOC, ELVOC, and LVOC, see SI). (C). $GR_{7-15}$ versus the predicted GR considering SA and condensable OOMs. The contribution of SA to particle growth is estimated using the equation by Stolzenburg et al.,(Stolzenburg et al., 2020) and the contribution of condensable OOM was calculated assuming ELVOC and LVOC were effectively non-volatile (Nieminen et al., 2010;Ehn et al., 2014). The measurement uncertainty (±50%) of SA(Kurten et al., 2012) is shown as the horizontal error bars. All plots are color-coded with the mean temperature at the corresponding time window. The linear correlation coefficients between the measured and calculated GRs in Panel B and C are also given.

## 4. Summary and Atmospheric implications

We examined the response of NPF to emission reductions in Beijing during the COVID-19 lockdown in both the molecular and the process levels. Clustering between SA and other base molecules drove the initial NPF in both pre-lockdown and lockdown periods. Our results show



that this clustering was insensitive to the emission reductions. However, it is evident that the
clustering efficiency of SA declined at high temperatures. This provides direct observational
evidence that traffic emissions alone cannot be a major source of NPF in Beijing, in contrast
to a few recent studies in urban areas (Ronkko et al., 2017; Guo et al., 2020).
The lockdown period showed an enhanced atmospheric oxidative capacity and reduced $SO_2$
concentrations; these balanced, so that both the SA concentration and particle formation rates
at 1.5 nm ($J_{1.5}$) were similar during the pre-lockdown and lockdown periods. This appears to
contradict a prior study reporting that NPF became stronger during the lockdown period based
on a measurement of particles down to 2 nm (Shen et al., 2021b). However, this apparent
discrepancy is mainly due to an increased particle survival probability caused by enhanced
particle growth during the lockdown period. To disentangle particle formation and growth,
measurement of particles at or below 1.5 nm is crucial to understanding the formation
mechanism of new particles.
The most obvious reason for the greater particle growth during the lockdown period was
elevated OOM concentrations due to enhanced photochemistry. We also expected that lower
$NO_x$ would favor particle growth, as NO can suppress particle growth by altering the OOM
composition and increasing the overall OOM volatility (Yan et al., 2020). This turned out not
to be the case in our study. We observed some changes in OOM composition in molecules
derived from oxidation of aromatic VOCs, but for the most part changes in OOM composition
and volatility were negligible. This suggests that the $RO_2$ + NO reaction remains important to
OOM formation even after such a dramatic $NO_x$ reduction. It has been proposed that
atmospheric $RO_2$ autoxidation will be increasingly more important if $NO_x$ keeps declining in
North America (Praske et al., 2018), which might potentially enhance peroxide-driven particle
toxicity and the yield of secondary organic aerosol (Zhao et al., 2017). However, our results
suggest that these adverse effects on human health and air quality are less likely to occur in
Beijing, at least in the near future.
A crucial challenge is to understand the key vapors and processes determining particle growth
rates. We investigated particle growth over three consecutive size ranges: 1.5 – 3 nm, 3 – 7 nm,
and 7 – 15 nm. Particle growth in each range shows distinct features and its relationship with
condensable vapors is the same in both periods. SA condensation almost completely explains
$GR_{1.5-3}$ at high temperatures. The co-condensation of nitric acid and ammonia might be an
important contributor at low temperatures, but this needs further verification by observations.
Condensation of OOMs plays a dominant role above 3 nm. Measured $GR_{3-7}$ and OOMs are
highly correlated. After scaling measured OOMs (by approximately a factor of 2) to account



for compounds that escape detection by $NO_3^-$ chemical ionization, the calculated $GR_{3-7}$ and $GR_{7-15}$ match the observed growth rates. The correlation with observations degrades at the larger size range, where particles are several hours old; there may be complex influences by other processes, such as the urban micro-meteorology and airmass inhomogeneity, which warrant future investigation.

## Acknowledgements

National Key R&D Program of China (2019YFC0214701, 2017YFC0209503, 2016YFC0200500), National Natural Science Foundation of China (41877306, 21876094) and Samsung PM2.5 SRP. All co-authors acknowledge the support of Beijing University of Chemical Technology. This work was supported by the Academy of Finland (1251427, 1139656, 296628, 306853, 316114, and 311932) & Finnish centre of excellence 1141135 & 307331, the EC Seventh Framework Program and European Union's Horizon 2020 program (ERC, project no.742206 "ATM-GTP", no. 850614 "CHAPAs"), the European Union's Horizon 2020 research and innovation programme under the Marie Sklodowska-Curie grant agreement No 895875 ("NPF-PANDA"), European Regional Development Fund, Urban innovative actions initiative (HOPE; Healthy Outdoor Premises for Everyone, project nro: UIA03-240), MegaSense by Business Finland (Grant 7517/31/2018), trans-national ERA-PLANET project SMURBS (Grant Agreement 689,443) under the EU Horizon 2020 Framework Programme, and Academy of Finland Flagship funding (grant no. 337549) is gratefully acknowledged. Centre of Excellence in Inverse Modeling and Imaging, Academy of Finland project 312125 is acknowledged. Kaspar R. Daellenbach received support from the Swiss National Science postdoc mobility grant (P2EZP2_18159). Juha Kangasluoma received funding from UHEL 3-year grant (75284132), Finnish Academy of Science project (1325656). Simo Hakala and Mona Kurppa acknowledge the doctoral programme in atmospheric sciences (ATM-DP, University of Helsinki). Neil Donahue acknowledges the US NSF (grant AGS1801897). Aijun Ding acknowledges the national natural science foundation of China (41725020). Lin Wang acknowledges the national natural science foundation of China (91644213, 21925601).

**Author contributions:** CY, YS, AD, JJ, and MK. designed the study; CY, YS, XQ, AMS, YG, LC, CD, ZL, and FZ conducted the measurement or collected key materials; CY, YS, XQ, LD, DS, SH, AMS, YG, TK, and JK analyzed the data; CY wrote the manuscript; all coauthors have read and commented on the manuscript.

**Competing interests:** The authors declare no competing interest.

**Data and materials availability:** Data and materials are available upon contacting the corresponding authors

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
