# Peer review of "The effect of COVID-19 restrictions on atmospheric new particle formation in Beijing"

_Atmospheric Chemistry and Physics, 2021_

## Author Comment (AC1)

**Response to comments on " The effect of COVID-19 restrictions on atmospheric new particle formation in Beijing"**

We thank the reviewers for their time, efforts, and constructive comments. We provide our point-to-point replies to these comments below. The comments by reviewers are in black, and the replies to the comments are in blue. The corresponding changes are noted in the manuscript and Supplementary Data with the same color code. All references are provided at the end of the replies.

**Referee 1**

There is a current debate on the driving mechanism(s) of NPF in urban environments, more specifically, sulfuric acid-amine clustering or oxidized organics originating from traffic emissions. The lockdown during COVID-19 pandemic provided a unique chance to definitively resolve this issue. Very simply put, if the strength of NPF reduced significantly during the lockdown, the dominant role of traffic emissions can be confirmed. A recent paper by Shen et al., (2021), that is also cited in this study, showed that NPF was stronger during the lockdown, which may suggest the less important role of traffic emissions in NPF in Beijing. In this study, the authors show consistent observational results with what have been reported by Shen et al., (2021), and further extended the mechanistic understanding of such NPF enhancement by performing detailed molecule-level analyses on NPF precursors, i.e., sulfuric acid and oxygenated organics. The authors found that the enhanced NPF were an overall result of two facts: first, the sulfuric acid-amine clustering remained as the driving mechanism and led to a similar $J_{1.5}$; second, the growth and survival of very small particles were enhanced by the elevated abundance of condensable oxidized organics.

Overall, I think this paper presents a significant advance in the understanding of NPF in urban environments, and thus I recommend accepting it for publication with a few minor comments/questions that I hope the authors can answer:

1. Line 216 - 218 "In addition, the concentration of OOMs increased by about 50% during the lockdown. This is because the concentration of volatile organic compounds (VOCs) only declined slightly in the lockdown period (Shen et al., 2021b), but the photochemistry was much more enhanced."

The overall pollution level was more serve during lockdown period. So will some of the OOMs be able to transport from other region(s) to the measurement site along with $PM_{2.5}$, and thus leading to the enhancement of OOM concentration?

Response: Thanks for the comments. The transport of OOMs depends on their atmospheric lifetimes. Firstly, OOMs with very low volatility are not expected to transport across long distances, because they would be lost on particles via condensation during the transport. In

Fig.R1, we show the volatility distribution of OOMs in this study, where ULVOCs, ELVOCs, LVOCs accounted for 5.2 %, 25.1 %, and 35.8 %, respectively. Secondly, the rest, relatively highly volatile OOMs (i.e., SVOCs, IVOCs, and VOCs) could partially survive from long-range transport. Yet, they may also lose via multi-generation of oxidations. Currently, limited information is available for the oxidation rates between these OOMs and oxidants, and therefore, it is hard to evaluate the fraction of OOMs lost via further oxidation during the transport. In summary, we can conclude that a minimum of ~ 2/3 of OOMs are formed locally. In addition, it is worth mentioning that the precursor of OOMs, i.e., various VOCs, are most likely transported from other regions. Therefore, the abundance of OOMs depends on both on the strength of photochemistry and air mass origins.

[Figure]

**Fig. R1.** Volatility distribution of total OOMs. ULVOC, ELVOC, LVOC, SVOC, IVOC and VOC denote ultra-low volatility organic compound, extremely low volatility organic compound, low volatility organic compound, semi-volatile organic compounds, intermediate volatility organic compound, and volatile organic compound respectively.

2. Line 285 "… range of our observations (Fig. S6)." Is this Fig. S6 should be Fig. S5?
Thanks. This has been corrected.
3. Line 291 "periods (Fig. S7). This is less than …" Is this Fig. S7 should be Fig. S6?
Thanks. This has been corrected.
4. Line 334-336 "When the NO concentration declined from the pre-lockdown period to the lockdown period, the ratio of $C_{6-9}H_{7,9,11,13}O_6N$ concentration to $C_{6-9}H_{7,9,11,13}O_5$ concentration decreased as well." Will the photolysis of nitrogen-containing aromatic OOMs influence the ratio

of $C_{6-9}H_{7,9,11,13}O_6N$ concentration to $C_{6-9}H_{7,9,11,13}O_5$ Concentration? And what will happen if color Fig. 5 (A) with UVB?

Response: Thanks for the comment. Photolysis is a loss pathway of some organic nitrates. As suggested, we colored Fig. 5 by UVB and the results are shown in Fig. R2.

It can be found that in all four panels, data points with different UVB values are well mixed, showing no clear dependence on the UVB level. This suggests that the UVB in wintertime Beijing is not a controlling factor that can substantially influence the ratio between nitrogen-containing and nitrogen-free OOMs.

[Figure]

**Fig. R2.** The influence of UVB (given by symbol color) on the composition of OOMs, indicated by the ratio between nitrogen-containing and nitrogen-free OOMs. (A) Selected OOMs with a double-bond-equivalent (DBE) of 3, which are usually products from the oxidation of aromatic compounds (Molteni et al., 2018; Wang et al., 2017; Garmash et al., 2020). (B) Selected OOMs

with a DBE of 1, which are more likely formed from the oxidation of aliphatic compounds, such as alkenes and alkanes. (C) OOMs containing 0 (CHO) and 1 ($CHON_1$) nitrogen atom. (D) OOMs containing 1 and 2 ($CHON_2$) nitrogen atoms. In all panels, only daytime data (7:00 – 18:00) with UVB higher than 0.001 W m$^{-2}$ were included. Circles and triangles represent data in pre-lockdown and lockdown periods, respectively.

5. Line 340 – 342 "They have a double bond equivalent (DBE) of 1, suggesting that they originate from aliphatic rather than aromatic precursors"
It seems that the authors have some criteria to infer the VOC precursor of OOMs. Is this based on some published results? I would like the authors to reply with more details.
Response: Yes, this is based on our previous studies (Nie et al., 2022; Guo et al., 2022). Nie et al. (2022) developed a novel workflow to retrieve the sources of OOMs. This workflow takes into account the composition of precursors and the current knowledge of atmospheric reaction pathways, which uses carbon number, oxygen number, nitrogen number, and the value of double-bond-equivalence (DBE) for the classification. This workflow classifies all OOMs with DBE≤1 and some OOMs with DBE=2 as aliphatic OOMs. This classification is also justified by various laboratory studies: neither monoterpenes (Ehn et al., 2012; Jokinen et al., 2014; Praplan et al., 2015; Boyd et al., 2015; Berndt et al., 2016) nor aromatics (Molteni et al., 2018; Wang et al., 2017; Garmash et al., 2020) could produce OOMs with DBE≤1. Thus, precursor VOCs with low DBE values are reasonable candidates, such as alkanes and alkenes.

**References**
Berndt, T., Richters, S., Jokinen, T., Hyttinen, N., Kurtén, T., Otkjær, R. V., Kjaergaard, H. G., Stratmann, F., Herrmann, H., Sipilä, M., Kulmala, M., and Ehn, M.: Hydroxyl radical-induced formation of highly oxidized organic compounds, Nature Communications, 7, 13677, 10.1038/ncomms13677, 2016.
Boyd, C. M., Sanchez, J., Xu, L., Eugene, A. J., Nah, T., Tuet, W. Y., Guzman, M. I., and Ng, N. L.: Secondary organic aerosol formation from the beta-pinene+NO3 system: effect of humidity and peroxy radical fate, Atmospheric Chemistry and Physics, 15, 7497-7522, 10.5194/acp-15-7497-2015, 2015.
Ehn, M., Kleist, E., Junninen, H., Petäjä, T., Lönn, G., Schobesberger, S., Dal Maso, M., Trimborn, A., Kulmala, M., Worsnop, D. R., Wahner, A., Wildt, J., and Mentel, T. F.: Gas phase formation of extremely oxidized pinene reaction products in chamber and ambient air, Atmospheric Chemistry and Physics, 12, 5113-5127, 10.5194/acp-12-5113-2012, 2012.
Garmash, O., Rissanen, M. P., Pullinen, I., Schmitt, S., Kausiala, O., Tillmann, R., Zhao, D., Percival, C., Bannan, T. J., Priestley, M., Hallquist, A. M., Kleist, E., Kiendler-Scharr, A., Hallquist, M., Berndt, T., McFiggans, G., Wildt, J., Mentel, T., and Ehn, M.: Multi-generation OH oxidation as a source for highly oxygenated organic molecules from aromatics, Atmospheric Chemistry and Physics, 20, 515-537, 10.5194/acp-20-515-2020, 2020.

Guo, Y., Yan, C., Liu, Y., Qiao, X., Zheng, F., Zhang, Y., Zhou, Y., Li, C., Fan, X., Lin, Z., Feng, Z., Zhang, Y., Zheng, P., Tian, L., Nie, W., Wang, Z., Huang, D., Daellenbach, K. R., Yao, L., Dada, L., Bianchi, F., Jiang, J., Liu, Y., Kerminen, V. M., and Kulmala, M.: Seasonal Variation of Oxygenated Organic Molecules in Urban Beijing and their Contribution to Secondary Organic Aerosol, Atmos. Chem. Phys. Discuss., 2022, 1-33, 10.5194/acp-2022-181, 2022.

Jokinen, T., Sipilä, M., Richters, S., Kerminen, V.-M., Paasonen, P., Stratmann, F., Worsnop, D., Kulmala, M., Ehn, M., Herrmann, H., and Berndt, T.: Rapid Autoxidation Forms Highly Oxidized RO2 Radicals in the Atmosphere, Angewandte Chemie International Edition, 53, 14596-14600, https://doi.org/10.1002/anie.201408566, 2014.

Molteni, U., Bianchi, F., Klein, F., El Haddad, I., Frege, C., Rossi, M. J., Dommen, J., and Baltensperger, U.: Formation of highly oxygenated organic molecules from aromatic compounds, Atmospheric Chemistry and Physics, 18, 1909-1921, 10.5194/acp-18-1909-2018, 2018.

Nie, W., Yan, C., Huang, D. D., Wang, Z., Liu, Y., Qiao, X., Guo, Y., Tian, L., Zheng, P., Xu, Z., Li, Y., Xu, Z., Qi, X., Sun, P., Wang, J., Zheng, F., Li, X., Yin, R., Dallenbach, K. R., Bianchi, F., Petäjä, T., Zhang, Y., Wang, M., Schervish, M., Wang, S., Qiao, L., Wang, Q., Zhou, M., Wang, H., Yu, C., Yao, D., Guo, H., Ye, P., Lee, S., Li, Y. J., Liu, Y., Chi, X., Kerminen, V.-M., Ehn, M., Donahue, N. M., Wang, T., Huang, C., Kulmala, M., Worsnop, D., Jiang, J., and Ding, A.: Secondary organic aerosol formed by condensing anthropogenic vapours over China's megacities, Nature Geoscience, 15, 255-261, 10.1038/s41561-022-00922-5, 2022.

Praplan, A. P., Schobesberger, S., Bianchi, F., Rissanen, M. P., Ehn, M., Jokinen, T., Junninen, H., Adamov, A., Amorim, A., Dommen, J., Duplissy, J., Hakala, J., Hansel, A., Heinritzi, M., Kangasluoma, J., Kirkby, J., Krapf, M., Kürten, A., Lehtipalo, K., Riccobono, F., Rondo, L., Sarnela, N., Simon, M., Tomé, A., Tröstl, J., Winkler, P. M., Williamson, C., Ye, P., Curtius, J., Baltensperger, U., Donahue, N. M., Kulmala, M., and Worsnop, D. R.: Elemental composition and clustering behaviour of α-pinene oxidation products for different oxidation conditions, Atmos. Chem. Phys., 15, 4145-4159, 10.5194/acp-15-4145-2015, 2015.

Wang, S., Wu, R., Berndt, T., Ehn, M., and Wang, L.: Formation of Highly Oxidized Radicals and Multifunctional Products from the Atmospheric Oxidation of Alkylbenzenes, Environmental Science & Technology, 51, 8442-8449, 10.1021/acs.est.7b02374, 2017.

---

## Author Comment (AC2)

**Response to comments on " The effect of COVID-19 restrictions on atmospheric new particle formation in Beijing"**

We thank the reviewers for their time, efforts, and constructive comments. We provide our point-to-point replies to these comments below. The comments by reviewers are in black, and the replies to the comments are in green. The corresponding changes are noted in the manuscript and Supplementary Data with the same color code. All references are provided at the end of the replies.

Reviewer #2

This paper presents the effect of the COVID-19 lockdown on atmospheric new particle formation. Indeed, the COVID-19 lockdown provided us a unique opportunity to investigate the effect of reduced anthropogenic emissions (probably similar to pre-industrial conditions) on a variety of atmospheric processes. Shen et al (2021) recently reported enhanced nanoparticle formation and growth during the COVID-19 lockdown in urban Beijing, but without much of the process-level explanation of nanoparticle formation and the role of key vapors. Here, the authors provide a more detailed analysis of nano particles and the role of sulfuric acid and oxygenated organic molecules in particle formation and growth. Authors report that the formation rate of 1.5 nm clusters was unchanged by drastically reduced traffic emissions. However, the cluster's survival probability was increased due to the higher formation of sulfuric acid, oxygenated organic molecules, and other vapors, indicating the enhanced atmospheric oxidative capacity.

Authors conclude that traffic emissions play a limited role in atmospheric NPF as opposed to the previous reports showing traffic as a high source of ultrafine particles such as Rönkkö et al., 2017 (https://doi.org/10.1073/pnas.1700830114), Guo et al., 2020 (https://doi.org/10.1073/pnas.1916366117). While Okuljar et al., 2021 (https://doi.org/10.5194/acp-21-9931-2021) also showed that traffic contribution to sub-3nm particles is lower during NPF events, Gani et al., 2021 (10.1039/D1EA00058F) showed NPF contributions to ultrafine particles in locations with high concentrations of precursors (e.g. traffic) are critical. Another recent study from an Indian urban location Kanawade et al., 2022 (https://doi.org/10.1029/2021JD035392), however, showed that NPF and growth events were suppressed under the reduced anthropogenic emissions during the lockdown. Kanawade et al. also reported an unaltered particle formation rate of 1.5 nm (and number concentrations of sub-3nm particles), but nanoparticle growth was limited by likely lower condensable vapors. This probably hints the role of micro-meteorology is also imperative. I suggest authors discussing all the above papers.

We thank the reviewer for this constructive comment. Indeed, several recent studies have suggested the contribution of traffic emissions to the concentration of 1-3 nm particles, as brought up by the reviewers. These studies were conducted in different locations and

environments, and because of this, the reported contribution of traffic emissions differs significantly. A mechanistic understanding of NPF, however, was not reached in these studies, due to either the lack of measurement of the NPF precursors (such as H2SO4, amines, NH3, and organic vapors) and/or the analysis of cluster dynamics vs. particle nucleation rate. Therefore, it is not so straightforward to compare our results to these studies.

For example, Gani et al., (2021) reported the contribution of traffic emissions to the concentration of ultrafine particles ($d_p$<100nm). In fact, this is consistent with our observations that a burst of ca. 7-30 nm particles can be observed during traffic rush hours at our measurement site (Fig. S3). However, we focus mainly on sub-3 nm particles in the context of particle nucleation, so the scope of these two studies is not entirely the same. Guo et al., (2020) suggested that organic vapors, as the oxidation products of traffic exhaust, are solely important for urban particle nucleation. This study was based on a chamber study, and without the measurement of NPF precursors, it remains unclear how well the chamber condition mimicked the ambient atmosphere. For these reasons, we feel that it is difficult to discuss the similarities and contrasts between our study and these two publications.

[Figure]

**Figure S3**. Particle number size distribution in NPF and non-event days during the pre-lockdown and lockdown periods.

On the other hand, other studies are relevant to this study (Ronkko et al., 2017; Okuljar et al., 2021; Kanawade et al., 2022). We added the following discussion to our manuscript (line 311-334) and updated the references.

"Ronkko et al., (2017) and Okuljar et al., (2021) both showed that in traffic-dense areas, the concentration of sub-3nm particles is obviously higher than in background areas. Kanawade et

al., (2022) conducted measurement of sub-3nm particles at a site that is ~ 1km away from traffic emission and found an insignificant influence of traffic emission on the particle concentration. These studies suggest that the distance between the measurement site and the traffic emission source is crucial for the observation of the emitted sub-3 nm particles, likely due to the dilution and coagulation loss of these nano-particles. However, it is probably not the same reason for our study, because the measurement site of this study is very close to an arterial road with heavy traffic. One possibility of the discrepancy is that the emission factor of sub-3nm particles is significantly lower for vehicles in Beijing. As shown in the laboratory study by Ronkko et al., (2017), the emission factor can vary by up to three orders of magnitude, being the highest for heavy-duty vehicles (e.g., diesel vehicles) and the lowest for light-duty cars. In Beijing, diesel vehicles are forbidden in downtown areas during traffic rush hours, so it is likely that the emission of sub-3nm particles is weak. Also, the high coagulation sink in Beijing and India might be another reason for the small contribution of traffic emissions. Another possibility that cannot be fully ruled out is the potential biases due to different detection methods of sub-3nm particles. The aforementioned studies utilized the PSM to detect sub-3nm particles, for which the size-classification of particles is based on the saturation ratio of diethylene glycol (DEG), while we use the soft Xray neutralizer and a DMA to classify particle size. The intrinsic difference between these two methods is not well quantified. It is also possible that the sub-3nm particles by vehicles are not efficiently charged by the soft Xray, and/or can be more efficiently activated by highly saturated DEG. Future research on the comparison between the PSM and SMPS is highly desired."

Overall Recommendation: The paper presents detailed analyses using new techniques that can characterize nanoparticles and provide new insights into the response of NPF to drastic changes in the atmospheric chemical cocktail. The manuscript should be published after the authors' elaborate discussion as indicated above and the following minor issues are addressed.

The pre-lockdown period falls during the peak winter season, followed by the lockdown during early spring, the temperature is expected to increase as the season progresses. The role of different micro-meteorological conditions should be highlighted between the time periods considered in this study. Or is it the critical factor for more occurrence of NPF and growth during lockdown with elevated temperature (more active photochemistry) rather than reduced anthropogenic emissions as background concentrations are on the higher side in urban areas.

Thanks for the comment. Indeed, the changes in temperature and solar radiation have multiple influences on particle formation and growth, which have been demonstrated in the manuscript. The increased UV radiation and atmospheric oxidative capacity (Fig. S4) outset the decreased SO2 concentration (Fig.2C), which results in a similar H2SO4 concentration (Fig. 2E). This is explained in the manuscript (Line 208-210 of the revised manuscript): "The median $SA_1$ and $SA_2$ concentrations were also stable between the two periods. This is because the decline of the

sulfuric acid precursor (i.e., $SO_2$, Figure 2C) was completely compensated by the enhanced photochemistry, as indicated by the variation of UVB (Fig. S4B)."

If the lockdown had not been imposed and $SO_2$ was not reduced, we would expect a higher $SA_1$ concentration in Feb than in Jan, due to the stronger UV radiation. However, a higher $SA_1$ concentration does not necessarily cause more frequent or stronger NPF. We found that the non-NPF days have a higher $SA_1$ concentration than that of NPF days due to the higher $SO_2$ concentration (Yan et al., 2021).

Our previous studies have shown that the CS is the governing parameter of the occurrence of NPF (Deng et al., 2021). Thus, the frequency of NPF in Beijing is primarily influenced by the origin of air masses. If the air mass came from the clean north area, the low CS due to the low concentration of pre-existing particles would cause more NPF events; and oppositely, if the air mass came from the polluted south area, there would be fewer NPF events.

Besides, high temperature can reduce the stability of SA clusters, as we show in Fig.4A, and ultimately weaken the strength of NPF (Deng et al., 2020). The high temperature should facilitate the formation of highly oxygenated organic molecules (HOM), but HOM are not the main precursor of initial particle formation (quantified by $J_{1.5}$), so this does not have a direct influence on the occurrence and strength of NPF.

Lines 85-90: there are laboratory studies showing clustering between sulfuric acid and organic acids e.g. Schobesberger et al. (https://doi.org/10.1073/pnas.130697311) or multi-component nucleation of sulfuric acid, ammonia, and organics (10.1126/sciadv.aau5363), and traffic is not the only source of organic acids to the atmosphere. For better readability, remove "on one hand" and "on other hand".

Agreed. We have removed "on one hand" and "on the other hand", as suggested.

Line 185: Fig. S4 cited for particles in the size range of 10-30 nm, but Fig. S4 in the supplementary shows diel patterns of temperature and UVB

Thanks for pointing out the error, Fig.S4 should be Fig.S3.

Lines 202-203: Correct as Fig. S4

Yes, this has been corrected.

Supplementary figures are incorrectly cited in the main text at most places. Please check carefully.

Yes, these have been checked and corrected.

Line 292: you mean to say "i.e., 1.3 pptv"?

Yes, we use "ppb" or "ppt" throughout the manuscript. To avoid confusion, we added "a volume mixing ratio of" in front of "1.3 ppt".

**References**

Kanawade, V. P., Sebastian, M., and Dasari, P.: Reduction in Anthropogenic Emissions Suppressed New Particle Formation and Growth: Insights From the COVID-19 Lockdown, Journal of Geophysical Research: Atmospheres, 127, e2021JD035392, 2022.

Okuljar, M., Kuuluvainen, H., Kontkanen, J., Garmash, O., Olin, M., Niemi, J. V., Timonen, H., Kangasluoma, J., Tham, Y. J., Baalbaki, R., Sipilä, M., Salo, L., Lintusaari, H., Portin, H., Teinilä, K., Aurela, M., Dal Maso, M., Rönkkö, T., Petäjä, T., and Paasonen, P.: Measurement report: The influence of traffic and new particle formation on the size distribution of 1–800 nm particles in Helsinki – a street canyon and an urban background station comparison, Atmos. Chem. Phys., 21, 9931-9953, 2021.

Ronkko, T., Kuuluvainen, H., Karjalainen, P., Keskinen, J., Hillamo, R., Niemi, J. V., Pirjola, L., Timonen, H. J., Saarikoski, S., Saukko, E., Jarvinen, A., Silvennoinen, H., Rostedt, A., Olin, M., Yli-Ojanpera, J., Nousiainen, P., Kousa, A., and Dal Maso, M.: Traffic is a major source of atmospheric nanocluster aerosol, Proc Natl Acad Sci U S A, 114, 7549-7554, 2017.

Yan, C., Yin, R., Lu, Y., Dada, L., Yang, D., Fu, Y., Kontkanen, J., Deng, C., Garmash, O., Ruan, J., Baalbaki, R., Schervish, M., Cai, R., Bloss, M., Chan, T., Chen, T., Chen, Q., Chen, X., Chen, Y., Chu, B., Dällenbach, K., Foreback, B., He, X., Heikkinen, L., Jokinen, T., Junninen, H., Kangasluoma, J., Kokkonen, T., Kurppa, M., Lehtipalo, K., Li, H., Li, H., Li, X., Liu, Y., Ma, Q., Paasonen, P., Rantala, P., Pileci, R. E., Rusanen, A., Sarnela, N., Simonen, P., Wang, S., Wang, W., Wang, Y., Xue, M., Yang, G., Yao, L., Zhou, Y., Kujansuu, J., Petäjä, T., Nie, W., Ma, Y., Ge, M., He, H., Donahue, N. M., Worsnop, D. R., Kerminen, V.-M., Wang, L., Liu, Y., Zheng, J., Kulmala, M., Jiang, J., and Bianchi, F.: The Synergistic Role of Sulfuric Acid, Bases, and Oxidized Organics Governing New-Particle Formation in Beijing, Geophysical Research Letters, 48, e2020GL091944, 2021.